# REPRESENTING ENTROPY : A SHORT PROOF OF THE EQUIVALENCE BETWEEN SOFT Q-LEARNING AND POLICY GRADIENTS

## ABSTRACT

Two main families of reinforcement learning algorithms, Q-learning and policy gradients, have recently been proven to be equivalent when using a softmax relaxation on one part, and an entropic regularization on the other. We relate this result to the well-known convex duality of Shannon entropy and the softmax function. Such a result is also known as the Donsker-Varadhan formula. This provides a short proof of the equivalence. We then interpret this duality further, and use ideas of convex analysis to prove a new policy inequality relative to soft Q-learning.

## 1 INTRODUCTION AND SETTING

Deep reinforcement learning as a research field is currently undergoing tremendous growth, largely due to empirical successes brought about by scaling the technique to real-world examples such as Atari games and Go. Historically, two main families of algorithms have existed:

- *Q-learning* (V. Mnih (2015)) proposes to iteratively refine estimates of a family of scalar *action-value* functions. These represent the reward expected after undertaking a given action, so as to be able to act greedily (or $\epsilon$-greedily) with respect to those numbers;

- *Policy gradients* (V. Mnih & Kavukcuoglu. (2016)), looks to maximize the expected reward by improving policies to favor high-reward actions. In general, the target loss function is regularized by the addition of an entropic functional for the policy. This makes policies more diffuse and less likely to yield degenerate results.

A critical step in the theoretical understanding of the field has been a smooth relaxation of the greedy max operation involved in selecting actions, turned into a Boltzmann softmax O. Nachum & Schuurmans. (2017b.). This new context has lead to a breakthrough this year J. Schulman & Abbeel. (2017) with the proof of the equivalence of both methods of Q-learning and policy gradients. While that result is extremely impressive in its unification, we argue that it is critical to look additionally at the fundamental reasons as to why it occurs. We believe that the convexity of the entropy functional used for policy regularization is at the root of the phenomenon, and that (Lagrangian) duality can be exploited as well, either yielding faster proofs, or further understanding. The contributions of our paper are as follows:

1. We show how convex duality expedites the proof of the equivalence between soft Q-learning and softmax entropic policy gradients - heuristically in the general case, rigorously in the bandit case.

2. We introduce a transportation inequality that relates the expected optimality gap of any policy with its Kullback-Leibler divergence to the optimal policy.

We describe our notations here. Abusing notation heavily by identifying measures with their densities as in $d\pi(a|s) = \pi(a|s)da$, if we note as either $r(s,a)$ or $r(a,s)$ the reward obtained by taking action $a$ in state $s$, the expected reward expands as:

$$K_r(\pi) = \mathbb{E}_\pi \big[ r(s,a) \big] = \int_{\mathbb{A}} r(s,a) d\pi(a|s) \tag{1}$$

$K_r$ is a linear functional of $\pi$. Adding Shannon entropic regularization[1] improves numerical stability of the algorithm, and prevents early convergence to degenerate solutions. Noting regularization strength $\beta$, the objective becomes a free energy functional, named by analogy with a similar quantity in statistical mechanics:

$$J(\pi) = \int_{\mathbb{A}} r(s,a)d\pi(a|s) - \beta \int_{\mathbb{A}} \log \pi(a|s)d\pi(a|s) \tag{2}$$

Crucially, viewed as a functional of $\pi$, $J$ is convex and is the sum of two parts

$$J(\pi) = K_r(\pi) - \beta H(\pi), \quad H(\pi) = \int_{\mathbb{A}} \log \pi(a|s)d\pi(a|s) \tag{3}$$

## 2 THE GIBBS VARIATIONAL PRINCIPLE FOR POLICY EVALUATION

### 2.1 LEGENDRE TRANSFORM AND POLICY ENTROPY

Here we are interested in the optimal *value* of the policy functional $J$, achieved for an optimal policy $\pi^*$. We hence look for $J^* = J(\pi^*) = \sup_{\pi \in \mathbb{P}} J(\pi)$. In the one step-one state bandit setting we are in, this is in fact almost the same as deriving the *state-value* function.

The principles of convex duality Bauschke & Combettes. (2011); Ziebart. (2010); G. Neu & Jonsson. (2017) yield a useful representation. Non-regularized empirical rewards in equation 1 can be seen as the standard inner product in Hilbert space $L^2$. We therefore equate inner product, expectation and integral over $\mathbb{A}$. Writing $J^*$ as

$$J^* = \sup_{\pi \in \mathbb{P}} J(\pi) = \sup_{\pi \in \mathbb{P}} \quad \langle r(s,a), \pi(a|s) \rangle - \beta H(\pi) \tag{4}$$

with $H$ the entropy functional defined above, **we recover exactly the definition of the Legendre-Fenchel transformation, or convex conjugate, of** $\beta \cdot H$. The word *convex* applies to the entropy functional, and doesn't make any assumptions on the rewards $r(s,a)$, other that they be well-behaved enough to be integrable in $a$.

The Legendre transform inverts derivatives. A simple calculation shows that the formal convex conjugate of $f : t \to t \log t$ is $f^* : p \to e^{(p-1)}$ - this because their respective derivatives $\log$ and $\exp$ are reciprocal. We can apply this to $f(\pi(a|s)) = \pi(a|s) \log \pi(a|s)$, and then this relationship can also be integrated in $a$. Hence the dual Legendre representation of the entropy functional $H$ is known. The Gibbs variational principle states that, taking $\beta = 1/\lambda$ as the inverse temperature parameter, and for each Borelian (measurable) test function $\Phi \in C^b(\mathbb{A})$:

$$\forall \Phi \in C^b(\mathbb{A}), \quad \sup_{\pi \in \mathbb{P}} \left[ \int_{\mathbb{A}} \Phi d\pi - \frac{1}{\lambda} H(\pi) \right] = \frac{1}{\lambda} \log \int_{\mathbb{A}} e^{\lambda \Phi} da \tag{5}$$

or in shorter notation, for each real random variable $X$ with exponential moments,

$$\forall X \in \mathbb{P}, \quad \sup_{\pi \in \mathbb{P}} \quad \mathbb{E}_\pi(X) - \frac{1}{\lambda} H(\pi) = \frac{1}{\lambda} \log \mathbb{E}(e^{\lambda X}) \tag{6}$$

We can prove a stronger result. If $\mu$ is a reference measure (or policy), and we now consider the *relative* entropy (or Kullback-Leibler divergence) with respect to $\mu$, $H_\mu(\cdot)$, instead of the entropy $H(\cdot)$, then the Gibbs variational principle still holds (Villani. (2008), chapter 22). This result regarding dual representation formulas for entropy is important and in fact found in several areas of science:

- as above, in thermodynamics, where it is named the *Gibbs variational principle*;
- in large deviations, this also known as the *Donsker-Varadhan* variational formula Dembo & Zeitouni. (2010);

---

[1]In this article we follow the convention of convex analysis, that is, entropy $H$ is taken to be convex, rather than that of information theory with H preceded by a negative sign and concave.

- in statistics, it is the well-known duality between maximum entropy and maximum likelihood estimation Altun & Smola. (2006);
- finally, the theory of information geometry Amari. (2016) groups all three views and posits that there exists a general, dually flat Riemannian information manifold.

The general form of the result is as follows. For each $\Phi$ representing a rewards function $r(s, a)$ or an estimator of it:

$$\forall \Phi \in C^b(\mathbb{A}), \quad \sup_{\pi \in \mathbb{P}} \Big[ \int_{\mathbb{A}} \Phi d\pi - \frac{1}{\lambda} H_\mu(\pi) \Big] = \frac{1}{\lambda} \log \int_{\mathbb{A}} e^{\lambda \Phi} d\mu \tag{7}$$

and the supremum is reached for the measure $\pi^* \in \mathbb{P}$ defined by its Radon-Nikodym derivative equal to the *Gibbs-Boltzmann measure* yielding an *energy* policy:

$$\frac{d\pi^*}{d\mu} = \frac{1}{Z} e^\Phi \tag{8}$$

In the special case where $\mu$ is the Lebesgue measure on a bounded domain (that is, the uniform policy), we find back the result 5 above, up to a constant irrelevant for maximization. In the general case, the mathematically inclined reader will also see this as a rephrasing of the fact the *Bregman divergence* associated with Shannon entropy is the Kullback-Leibler divergence. For completeness' sake, we provide here its full proof :

**Proposition 1.** ***Donsker-Varadhan variational formula***. *Let $G$ be a bounded measurable function on $\mathcal{A}$ and $\pi$, $\tilde{\pi}$ be probability measures on $\mathcal{A}$, with $\pi$ absolutely continuous w.r.t. $\tilde{\pi}$. Then*

$$\int_{\mathcal{A}} G d\pi - \tau D_{\mathrm{KL}}[\pi \| \tilde{\pi}] = \ln \int_{\mathcal{A}} e^{G/\tau} d\tilde{\pi} - \tau D_{\mathrm{KL}}[\pi \| \pi^*] \tag{9}$$

*where $\pi^*$ is a probability measure defined by the Radon-Nikodym derivative:*

$$\frac{d\pi^*}{d\tilde{\pi}} = \frac{e^{G/\tau}}{\int_{\mathcal{A}} e^{G/\tau} d\tilde{\pi}} \tag{10}$$

*Proof.*

$$
\begin{aligned}
\int_{\mathcal{A}} G d\pi - \tau D_{\mathrm{KL}}[\pi \| \tilde{\pi}] &= \int_{\mathcal{A}} G d\pi - \tau \int_{\mathcal{A}} \big( \ln \frac{d\pi}{d\tilde{\pi}} \big) d\pi \\
&= \int_{\mathcal{A}} G d\pi - \tau \int_{\mathcal{A}} \big( \ln \frac{d\pi}{d\pi^*} \big) d\pi - \tau \int_{\mathcal{A}} \big( \ln \frac{d\pi^*}{d\tilde{\pi}} \big) d\pi \\
&= \int_{\mathcal{A}} \Big( G - \tau \big( \ln \frac{d\pi^*}{d\tilde{\pi}} \big) \Big) d\pi - \tau D_{\mathrm{KL}}[\pi \| \pi^*] \\
&= \int_{\mathcal{A}} \Big( G - \tau \big( \ln \frac{e^{G/\tau}}{\int_{\mathcal{A}} e^{G/\tau} d\tilde{\pi}} \big) \Big) d\pi - \tau D_{\mathrm{KL}}[\pi \| \pi^*] \\
&= \int_{\mathcal{A}} \big( \ln \int_{\mathcal{A}} e^{G/\tau} d\tilde{\pi} \big) d\pi - \tau D_{\mathrm{KL}}[\pi \| \pi^*] \\
&= \ln \int_{\mathcal{A}} e^{G/\tau} d\tilde{\pi} - \tau D_{\mathrm{KL}}[\pi \| \pi^*]
\end{aligned}
$$

$\square$

**Proposition 2.** *Corollary :*

$$\max_{\pi} \Big[ \int_{\mathcal{A}} G d\pi - \tau D_{\mathrm{KL}}[\pi \| \tilde{\pi}] \Big] = \ln \int_{\mathcal{A}} e^{G/\tau} d\tilde{\pi} \tag{11}$$

*and the maximum is attained uniquely by $\pi^*$.*

*Proof.* $D_{\mathrm{KL}}[\pi \| \pi^*] \geq 0$, and $D_{\mathrm{KL}}[\pi \| \pi^*] = 0$ if and only if $\pi = \pi^*$. $\square$

The link with reinforcement learning is made by picking $\Phi = r(s, a)$, $\pi = \pi(a|s)$, $\lambda = 1/\beta$, and by recalling the implicit dependency of the right member on $s$ but not on $\pi$ at optimality, so that we can write

$$J^* = V^*(s) = \beta \cdot \log \int_{\mathbb{A}} e^{r(s,a)/\beta} d\mu(a) \tag{12}$$

which is the definition of the one-step soft Bellman operator at optimum R. Fox & Tishby. (2015); O. Nachum & Schuurmans. (2017b.); T. Haarnoja & Levine. (2017). Note that here $V^*(s)$ depends on the reference measure $\mu$ which is used to pick actions frequency - we can be off-policy, in which case $V^*$ is only a pseudo state-value function.

## 2.2 PROVING SOFT Q-LEARNING EQUIVALENCE

In this simplified one-step setting, this provides a short and direct proof that *in expectation, and trained to optimality,* soft Q-learning and policy gradients ascent yield the same result J. Schulman & Abbeel. (2017). Standard Q-learning is the special case $\beta \rightarrow 0$, $\lambda \rightarrow \infty$ where by the Laplace principle we recover $V(s) \rightarrow \max_{\mathbb{A}} r(s, a)$ ; that is, the zero-temperature limit, with no entropy regularization. For simplicity of exposition, we have restricted so far to the proof in the bandit setting; now we extend it to the general case.

First by inserting $V^*(s) = \sup_{\pi} V^{\pi}(s)$ in the representation formulas above applied to $r(s, a) + \gamma V^*(s')$, so that

$$V^*(s) = \sup_{\pi} \left[ \mathbb{E}_{\pi}[r(s, a) + \gamma V^*(s')] - \beta H(\pi) \right] = \beta \cdot \log \int_{\mathbb{A}} e^{\frac{r(s,a) + \gamma V^*(s')}{\beta}} da \tag{13}$$

The proof in the general case will then be finished if we assume that we could apply the Bellman optimality principle not to the hard-max, but to the soft-max operator. This requires proving that the soft-Bellman operator admits a unique fixed point, which is the above. By the Brouwer fixed point theorem, it is enough to prove that it is a contraction, or at least non-expansive (we assume that the discount factor $\gamma < 1$ to that end). We do so below, noting that this result has been shown many times in the literature, for instance in O. Nachum & Schuurmans. (2017b.). Refining the soft-Bellman operator just like above, but in the multi-step case, by the expression

$$(\mathcal{B}^* V)(s) = \beta \cdot \log \int_a e^{\frac{r(s,a) + \gamma \mathbb{E}_{s'|s,a}(V(s'))}{\beta}} da \tag{14}$$

we get the:

**Proposition 3.** *Nonexpansiveness of the soft-Bellman operator for the supremum norm $\|f\|_{\infty}$.*

$$\left\| \mathcal{B}^* V^{(1)} - \mathcal{B}^* V^{(2)} \right\|_{\infty} < \|V^{(1)} - V^{(2)}\|_{\infty} \tag{15}$$

*Proof.* Let us consider two state-value functions $V^{(1)}(s)$ and $V^{(2)}(s)$ along with the associated action-value functions $Q^{(1)}(s, a)$ and $Q^{(2)}(s, a)$. Besides, denote MDP transition probability by $p(s'|s, a)$. Then :

$$
\begin{aligned}
\left\| \mathcal{B}^* V^{(1)} - \mathcal{B}^* V^{(2)} \right\|_{\infty} &= \max_s \left| (\mathcal{B}^* V^{(1)})(s) - (\mathcal{B}^* V^{(2)})(s) \right| \\
&\leq \max_s \max_a \left| Q^{(1)}(s, a) - Q^{(2)}(s, a) \right| \\
&= \gamma \max_s \max_a \left| \mathbb{E}_{s'|s,a} \left[ V^{(1)}(s') - V^{(2)}(s') \right] \right| \\
&\leq \gamma \max_s \max_a \|p(s'|s, a)\|_1 \|V^{(1)} - V^{(2)}\|_{\infty} \quad \text{by Hölder's inequality} \\
&= \gamma \|V^{(1)} - V^{(2)}\|_{\infty} < \|V^{(1)} - V^{(2)}\|_{\infty}
\end{aligned}
$$

$\square$

## 2.3 INTERPRETATION

In summary, the program of the proof was as below :

1. Write down the entropy-regularised policy gradient functional, and apply the Donsker-Varadhan formula to it.

2. Write down the resulting softmax Bellman operator as a solution to the sup maximization - this obviously also proves existence.

3. Show that the softmax operator, just like the hard max, is still a contraction for the max norm, hence prove uniqueness of the solution by fixed point theorem.

The above also shows formally that, should we discretize the action space $\mathbb{A}$ to replace integration over actions by finite sums, any strong estimator $\hat{r}(s, a)$ of $r(s, a)$, applied to the partition function of rewards $\frac{1}{\lambda} \log \sum_a e^{\lambda r(s,a)}$, could be used for Q-learning-like iterations. This is because strong convergence would imply weak convergence (especially convergence of the characteristic function, via Levy's continuity theorem), and hence convergence towards the log-sum-exp cumulant generative function above. Different estimators $\hat{r}(s, a)$ lead to different algorithms. When the MDP and the rewards function $r$ are not known, the parameterised critic choice $\hat{r}(s, a) \approx Q_w(s, a)$ recovers Nachum's Path Consistency Learning O. Nachum & Schuurmans. (2017b.;c). O'Donoghue's PGQ method B. O'Donoghue & Mnih. (2016) can be seen as a control variate balancing of the two terms in 7. In theory, the rewards distribution could be also recovered simply by varying $\lambda$ (or $\beta$), for instance by inverse Laplace transform.

## 3 POLICY OPTIMALITY GAP AND TEMPERATURE ANNEALING

In this section, we propose an inequality that relates the *optimality gap* of a policy - by how much that policy is sub-optimal on average - to the Kullback-Leibler divergence between the current policy and the optimum. The proof draws on ideas of convex analysis and Legendre transormation exposed earlier in the context of soft Q-learning.

Let us assume that $X$ is a real-valued *bounded* random variable. We denote $\sup |X| \leq M$ with $M$ constant. Furthermore we assume that $X$ is centered, that is, $\mathbb{E}[X] = 0$. This can always be achieved just by picking $Y = X - \mathbb{E}[X]$.

Then, by the Hoeffding inequality :

$$\log \mathbb{E}(e^{\beta X}) \leq K \frac{\beta^2}{2} \tag{16}$$

with $K$ a positive real constant, i.e., the variable $X$ is sub-Gaussian, so that its cumulant generating function grows less than quadratically. By taking a Legendre transformation and inverting it, we get that for any pair of measures $\mathbb{P}$ and $\mathbb{Q}$ that are mutually absolutely continuous, one has

$$\mathbb{E}_{\mathbb{Q}}(X) - \mathbb{E}_{\mathbb{P}}(X) \leq \sqrt{2K \cdot D_{KL}(\mathbb{Q}||\mathbb{P})} \tag{17}$$

which by specializing $\mathbb{Q}$ to be the measure associated to $\mathbb{P}^*$ the optimal policy, $\mathbb{P}_\theta$ the current parameterized policy, and $X$ an advantage return $r$ :

$$\mathbb{E}_{\mathbb{P}^*}(r) \leq \mathbb{E}_{\mathbb{P}_\theta}(r) + \sqrt{2K}\sqrt{D_{KL}(\mathbb{P}^*||\mathbb{P}_\theta)} \tag{18}$$

By the same logic, any upper bound on $\log \mathbb{E}(e^{\beta X})$ can give us information about $\mathbb{E}_{\mathbb{Q}}(X) - \mathbb{E}_{\mathbb{P}}(X)$. This enables us to relate the size of Kullback-Leibler trust regions to the amount by which our policy could be improved. In fact by combining the entropy duality formula with the Legendre transformation, one easily proves the below :

**Proposition 4.** *Let $X$ a real-valued integrable random variable, and $f$ a convex and differentiable function such that $f(0) = f'(0) = 0$. Then with $f^* : x \to f^*(x) = \sup(\beta x - f(\beta))$ the Legendre transformation of $f$, $f^{*-1}$ its reciprocal, and $\mathbb{P}$ and $\mathbb{Q}$ any two mutually absolutely continuous measures, one has the equivalence:*

$$\log \mathbb{E}_{\mathbb{P}}(e^{\beta(X - \mathbb{E}_{\mathbb{P}}(X))}) \leq f(\beta) \quad \Longleftrightarrow \quad \mathbb{E}_{\mathbb{Q}}(X) - \mathbb{E}_{\mathbb{P}}(X) \leq f^{*-1}[D_{KL}(\mathbb{Q}||\mathbb{P})] \tag{19}$$

*Proof.* By Donsker-Varadhan formula, one has that the equivalence is proven if and only if

$$\mathbb{E}_{\mathbb{Q}}(X) - \mathbb{E}_{\mathbb{P}}(X) \le \inf_{\beta}\Big[\frac{f(\beta) + D_{KL}(\mathbb{Q}||\mathbb{P})}{\beta}\Big] \tag{20}$$

but this right term is easily proven to be nothing but

$$f^{*-1}(D_{KL}(\mathbb{Q}||\mathbb{P})) \tag{21}$$

the inverse of the Legendre transformation of $f$ applied to $D_{KL}(\mathbb{Q}||\mathbb{P})$. □

This also opens up the possibility of using various softmax temperatures $\beta_i$ in practical algorithms in order to estimate $f$. Finally, note that if $\mathbb{P}_\theta$ is a parameterized softmax policy associated with action-value functions $Q_\theta(a, s)$ and temperature $\beta$, then because $\mathbb{P}^*$ is proportional to $e^{-r(a,s)/\beta}$, one readily has

$$D_{KL}\big(\mathbb{P}^*||\mathbb{P}_\theta\big) = \frac{1}{\beta}\Big[\mathbb{E}(Q_\theta) - \mathbb{E}(r)\Big] \tag{22}$$

which can easily be inserted in the inequality above for the special case $\mathbb{Q} = \mathbb{P}^*$.

## 4 RELATED WORK

Entropic reinforcement learning has appeared early in the literature with two different motivations. The view of exploration with a self-information intrinsic reward was pioneered by Tishby, and developed in Ziebart's PhD. thesis Ziebart. (2010). It was rediscovered recently that within the asynchronous actor-critic framework, entropic regularization is crucial to ensure convergence in practice V. Mnih & Kavukcuoglu. (2016). Furthermore, the idea of taking steepest KL divergence steps as a practical reinforcement learning method per se was adopted by Schulman J. Schulman & Abbeel. (2015a.). The Lagrangian duality view was pioneered in a practical context with O'Donoghue's PGQ algorithm B. O'Donoghue & Mnih. (2016), and followed by the development of soft Q-learning jointly in R. Fox & Tishby. (2015) and in Nachum et al. O. Nachum & Schuurmans. (2017b.). The key common development in these works has been to make entropic regularization recursively follow the Bellman equation, rather than naively regularizing one-step policies G. Neu & Jonsson. (2017). Schulman thereafter proposed a general proof of the equivalence, in the limit, of policy gradient and soft Q-learning methods J. Schulman & Abbeel. (2017), but the proof does not explicitly make the connection with convex duality and the expeditive justification it yields in the one-step case. Applying the Gibbs/Donsker-Varadhan variational formula to entropy in a machine learning context is, however, not new; see for instance Altun and Smola Altun & Smola. (2006). Some of the convex optimization results they invoke, including proximal stepping, can be found in the complete treatment by Bauschke Bauschke & Combettes. (2011). In the context of neural networks, convex analysis and partial differential equation methods are covered by Chaudhari P. Chaudhari & Carlier. (2017).

## 5 FURTHER WORK

Using dual formulas for the entropy functional in reinforcement learning has vast potential ramifications. One avenue of research will be to interpret our findings in a large deviations framework - the log-sum-exp cumulant generative function being an example of *rate function* governing fluctuations of the tail of empirical n-step returns. Smart drift change techniques could lead to significant variance reduction for Monte-Carlo rollout estimators. We also hope to exploit further concentration inequalities in order to provide more bounds for the state value function. Finally, a complete theory of the one-to-one correspondence between convex approximation algorithms and reinforcement learning methods is still lacking to date. We hope to be able to contribute in this direction through further work.

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
