# OpenReview forum: "Representing Entropy : A short proof of the equivalence between soft Q-learning and policy gradients"
_ICLR.cc/2018/Conference — Reject_

### Official Review · AnonReviewer1 · 2017-11-27
**main results are already known in the context of risk-sensitive control**

**Rating:** 5
**Confidence:** 5

**Review:**

This paper uses a well-known variational representation of the relative entropy (the so-called Donsker-Varadhan formula) to derive an expression for the Bellman error with entropy regularization in terms of a certain log-partition function. This is stated in Equation (13) in the paper. However, this precise representation of the Bellman error (with costs instead of rewards and with minimization instead of maximization) has been known in the literature on risk-sensitive control, see, e.g., P. D. Pra, L. Meneghini, and W. J. Runggaldier, “Connections between stochastic control and dynamic games,” Math. Control Signals Systems, vol. 9, pp. 303–326, 1996. The same applies to contraction results for the "softmax" Bellman operator -- these results are not novel at all, see, e.g., D. Hernandez-Hernandez and S. I. Marcus, “Risk sensitive control of Markov processes in countable state space,” Systems and Control Letters, vol. 29, pp. 147–155, 1996.

Also, there are some errors in the paper: for example, the functional of $\pi(a|s)$ in Eq. (2) is concave, not convex, since the expression for the Shannon entropy in Eq. (3) has the wrong sign.

---

> ### Author Response · Authors · 2018-01-05
> **RE : Risk sensitive control**
>
> We take due note of the fact that these duality results are already known in reinforcement learning - we were not aware of either 1996-dated paper, and want to respectfully thank the reviewer for their mention, and sharing their extensive knowledge of the field.
>
> The sign typo regarding the J functional is a valid point that has been corrected.

---

### Official Review · AnonReviewer3 · 2017-11-27
**Empty paper.**

**Rating:** 2
**Confidence:** 5

**Review:**

Summary
*******
The paper provides a collection of existing results in statistics.

Comments
********
Page 1: references to Q-learning and Policy-gradients look awkwardly recent, given that these have been around for several decades.

I dont get what is the novelty in this paper. There is no doubt that all the tools that are detailed here are extremely useful and powerful results in mathematical statistics. But they are all known.

The Gibbs variational principle is folklore, Proposition 1,2 are available in all good text books on the topic,
and Proposition 4 is nothing but a transportation Lemma.
Now, Proposition 3 is about soft-Bellman operators. This perhaps is less standard because contraction property of soft-Bellman operator in infinite norm is more recent than for Bellman operators.
But as mentioned by the authors, this is not new either.
Also I don't really see the point of providing the proofs of these results in the main material, and not for instance in appendix, as there is no novelty either in the proof techniques.

I don't get the sentence "we have restricted so far the proof in the bandit setting": bandits are not even mentioned earlier.

Decision
********
I am sorry but unless I missed something (that then should be clarified) this seems to be an empty paper: Strong reject.

---

> ### Author Response · Authors · 2018-01-05
> **The method of the proof**
>
> We thank the reviewer for their evaluation, and acknowledge it. However, we are not in full agreement with the specific concern voiced here - our shorter proof indeed uses fairly well-known statistical tools, for instance in the Legendre transform of the log-Laplace. But of the several papers highlighting the equivalence of soft Q-learning and entropy regularized policy gradients published this year (at least three, see for instance Nachum et al.'s https://arxiv.org/abs/1702.08892 or Schulman et al.'s https://arxiv.org/abs/1704.06440, or the anonymous submission https://openreview.net/pdf?id=HJjvxl-Cb), none to our knowledge used this representation formula that expedites the proof singularly. The technique gives very intuitively soft Q-learning as a Cramer-Chernoff transform, and could be applied to other regularizers ; furthermore, the paper highlights a connection with large deviations that could be helpful in future work, for instance by applying relevant changes of measure.
>
> The sentence 'we have restricted so far the proof in the bandit setting' refers to applying the representation formula in the one-step return case for clarity ; this is terminology used for instance in Schulman et al.'s article https://arxiv.org/abs/1704.06440.
>
> We do agree - and state in the abstract -  that the proof of equivalence of soft Q-learning and entropy regularization is not a novelty of our article.

---

### Official Review · AnonReviewer2 · 2017-12-01
**The paper presents a concise proof of the equivalence between entropy-regularised policy gradients and soft-max Q learning. It first shows the results in the bandit-case and extend it to the MDP case. The presentation is a little bit more general and related to works outside the field of machine learning. However, the results are not new and are a survey of previous works (e.g., Neu 2017).**

**Rating:** 5
**Confidence:** 4

**Review:**

Clarity: The paper is easy to follow and presents quite well the equivalence.

Originality: The results presented are well known and there is no clear contribution algorithmic-wise to the field of RL. The originality comes from the conciseness of the proof and how it relates to other works outside ML. Thus, this contribution seems minor and out of the scope of the conference which focus on representation learning for ML and RL.

Suggestion: I strongly suggest the authors to work on a more detailed proof for the RL case explaining for instance the minimal conditions (on the reward, on the ergodicity of the MDP) in which the equivalence holds and submit it to a more theoretically oriented conference such as COLT or NIPS.

---

> ### Author Response · Authors · 2018-01-05
> **Thank you for your suggestions !**
>
> We respectfully acknowledge the reviewer's comments, and will indeed endeavour to take a more theoretical angle on minimal conditions for the proof in further work. This is an extremely helpful suggestion. We are thankful for comments on clarity/writing style.
>
> We want to sincerely thank the reviewer for their insight and their time.

---

### Decision · Program_Chairs · 2018-01-29
**ICLR 2018 Conference Acceptance Decision**

**Decision:**

Reject

**Comment:**

The reviewers point out that this is a well known result and is not novel.